# Real-World Data on Chronic Myelomonocytic Leukemia: Clinical and Molecular Characteristics, Treatment, Emerging Drugs, and Patient Outcomes

**DOI:** 10.3390/cancers14174107

**Published:** 2022-08-25

**Authors:** Sandra Castaño-Díez, Mónica López-Guerra, Cristina Bosch-Castañeda, Alex Bataller, Paola Charry, Daniel Esteban, Francesca Guijarro, Carlos Jiménez-Vicente, Carlos Castillo-Girón, Albert Cortes, Alexandra Martínez-Roca, Ana Triguero, José Ramón Álamo, Silvia Beà, Dolors Costa, Dolors Colomer, María Rozman, Jordi Esteve, Marina Díaz-Beyá

**Affiliations:** 1Hematology and Hematopathology Departments, Hospital Clínic Barcelona, 08036 Barcelona, Spain; 2Medical School, University of Barcelona, 08036 Barcelona, Spain; 3August Pi i Sunyer Biomedical Research Institute (IDIBAPS), 08036 Barcelona, Spain; 4Centro de Investigación Biomédica en Red de Cáncer (CIBERONC), 28029 Madrid, Spain; 5Josep Carreras Leukemia Research Institute, 08916 Badalona, Spain; 6Hematology Department, Hospital de la Santa Creu i Sant Pau, 08025 Barcelona, Spain

**Keywords:** chronic myelomonocytic leukemia, CMML, AML transformation, clonal evolution, prognosis, treatment, hypomethylating agents, targeted therapy, gene mutations, survival

## Abstract

**Simple Summary:**

Chronic myelomonocytic leukemia (CMML) is an infrequent disease with poor prognosis and risk of progression into acute myeloid leukemia (AML). Stem cell transplantation (alloSCT) is the only potentially curative option. New targeted drugs (NTDs) directed at specific gene mutations are useful in AML, but little is known about how CMML progresses to AML and if these drugs are effective in CMML. In our study, 38% of patients received hypomethylating agents but less than half of them responded. Six patients received NTDs and responded well. AlloSCT was possible in only 10% of patients. Progression to AML occurred in 25% of patients, and there were changes in their gene mutations between the time of diagnosis and the time of progression. Although prognosis is poor in CMML, analyzing gene mutations can help to better stratify the risk of each patient and to identify potentially effective NTDs for each patient.

**Abstract:**

Despite emerging molecular information on chronic myelomonocytic leukemia (CMML), patient outcome remains unsatisfactory and little is known about the transformation to acute myeloid leukemia (AML). In a single-center cohort of 219 CMML patients, we explored the potential correlation between clinical features, gene mutations, and treatment regimens with overall survival (OS) and clonal evolution into AML. The most commonly detected mutations were *TET2*, *SRSF2*, *ASXL1*, and *RUNX1*. Median OS was 34 months and varied according to age, cytogenetic risk, FAB, CPSS and CPSS-Mol categories, and number of gene mutations. Hypomethylating agents were administered to 37 patients, 18 of whom responded. Allogeneic stem cell transplantation (alloSCT) was performed in 22 patients. Two-year OS after alloSCT was 60.6%. Six patients received targeted therapy with *IDH* or *FLT3* inhibitors, three of whom attained a long-lasting response. AML transformation occurred in 53 patients and the analysis of paired samples showed changes in gene mutation status. Our real-world data emphasize that the outcome of CMML patients is still unsatisfactory and alloSCT remains the only potentially curative treatment. However, targeted therapies show promise in patients with specific gene mutations. Complete molecular characterization can help to improve risk stratification, understand transformation, and personalize therapy.

## 1. Introduction

Chronic myelomonocytic leukemia (CMML) is a clonal hematopoietic stem cell disorder consisting of persistent monocytosis in peripheral blood (≥1 × 10^9^/L monocytes over 3 months), ≥10% of monocytes in the white blood cell count (WBC), dysplasia involving one or more myeloid lineages, and an acquired clonal cytogenetic or molecular abnormality in hematopoietic cells [1]. CMML is classified as a distinct entity in the myelodysplastic/myeloproliferative (MDS/MPN) category [1] and subdivided according to leukocyte cell count by the French-American-British (FAB) classification [2] and by blast and promonocyte percentage in blood and bone marrow by the World Health Organization (WHO) [1]. CMML is a rare disease in general but with an incidence of 2.5 cases per 100,000 individuals > 70 years [3]. Median age at CMML diagnosis is 72 years and almost 90% of cases occur in patients > 60 years [4,5]. Around 20–30% of patients with CMML have cytogenetic abnormalities at diagnosis [6] and almost 90% harbor at least one somatic mutation, most frequently of *TET2*, *SRSF2*, and *ASXL1* [7].

Median overall survival (OS) of CMML patients is 29 months [8,9], and about 25% of patients progress to acute myeloid leukemia (AML). The most frequently used risk stratification scales for CMML are the CMML Prognostic Scoring System (CPSS) and the CPSS-Molecular (CPSS-Mol). The CPSS includes the FAB classification, the WHO category, transfusion dependence, and cytogenetic risk, while the CPSS-Mol adds the negative prognostic value of *ASXL1*, *NRAS*, *SETBP1*, and/or *RUNX1* mutations. In approximately half of CMML patients, the CPSS-Mol calculates a higher estimate of individual risk than the CPSS [10]. Both scales recognize four risk groups (low, intermediate-1, intermediate-2, and high), each with a different OS and AML transformation rate [10,11]. In fact, the pathophysiology of AML transformation in CMML patients remains a challenge [12,13]. Sequential studies with paired samples of CMML at diagnosis and at the time of AML transformation [14,15,16] have suggested a key role for the *RAS* pathway, but there are still blind spots to be elucidated, and there is a crucial need for a better understanding of the molecular mechanisms involved in AML transformation in order to design effective targeted treatments.

In real-world clinical practice, patients are generally classified into two risk groups: low (low or intermediate-1 risk in CPSS or CPSS-Mol) or high (intermediate-2 or high CPSS or CPSS-Mol). Low-risk patients are generally treated with non-disease-modifying treatments, such as erythropoietin-stimulating agents for anemia and hydroxyurea for leukocytosis in patients with the MPN phenotype, while high-risk patients receive disease-modifying treatments, including hypomethylating agents (HMAs), chemotherapy, and allogeneic hematopoietic stem cell transplantation (alloSCT) [4]. AlloSCT is the only treatment option with proven curative capacity, although 5-year OS after alloSCT is still only 20–50% [17,18,19,20,21]. Moreover, in fact, due mainly to the older age and comorbidities of most CMML patients, alloSCT is only feasible in around 10% of patients [22,23].

In recent years, several targeted therapies have been used and approved for other myeloid neoplasms, including AML. For example, patients with *FLT3* mutations can be treated with *FLT3* inhibitors (midostaurin, gilteritinib, sorafenib), and those with *IDH1/2* mutations can receive *IDH1/2* inhibitors (ivosidenib, enasidenib). However, there is still very limited clearly defined data on the efficacy of these treatments in CMML [24,25,26,27] since CMML patients are commonly not included in clinical trials of MDS/AML, due both to the low incidence of CMML and the older age of the patients [24,28].

Given this scarcity of clinical trials in CMML, there is increasing interest in the use of real-world data to guide and improve disease management and to refine risk stratification. Here we report our retrospective study of clinical features, gene mutations, treatments (including novel targeted therapies), AML transformation, and patient outcome in a large cohort of CMML patients with long follow-up at a single center.

## 2. Materials and Methods

### 2.1. Patients

We included a consecutive series of 219 patients diagnosed with CMML at Hospital Clínic of Barcelona, Spain from 1997 to 2021. CMML was morphologically defined according to the 2017 FAB and WHO criteria [1,2]. Clinical and molecular data were collected from patient records. Cytogenetic and mutational analyses were performed during patient treatment and follow-up, and results were examined retrospectively for the present study.

### 2.2. Cytogenetics

Metaphase cytogenetic profiling was carried out on samples from bone marrow aspirates. Chromosomal preparation was performed on G-banded metaphase cells using standard techniques. Karyotypes were reported according to the International System for Human Cytogenetic Nomenclature [29], and cytogenetic risk was calculated according to the Spanish cytogenetic risk-stratification system: low risk (normal karyotype or loss of chromosome Y as a single abnormality); intermediate risk (all chromosomal abnormalities except those included in the low- and high-risk categories); and high risk (trisomy 8, chromosome 7 abnormalities or complex karyotype [≥3 chromosomal abnormalities]) [9].

### 2.3. Gene Mutations

Bone marrow samples were collected at the time of CMML diagnosis and, when possible, at the time of AML transformation. Targeted next generation sequencing (NGS) was carried out on genomic DNA using Ion Ampliseq^TM^ AML Research Panel (*n* = 17) or Oncomine^TM^ Myeloid Research Assay panel (ThermoFisher Scientific, Waltham, MA, USA) (*n* = 78) (Appendix A). Sequencing data were analyzed using the Ion Reporter software (ThermoFisher Scientific). The minimum VAF for calling single nucleotide variants was 1%. Synonymous, intronic, and polymorphic variants were ruled out. Selected variants were reviewed in COSMIC (Catalog of Somatic Mutations in Cancer), ClinVar, and Varsome databases to finally select the clinically relevant variants. Molecular screening of *FLT3*-ITD, *FLT3*-TKD, and *NPM1* mutations was performed by conventional PCR-based techniques and the *FLT3*-ITD allelic ratio was estimated using PCR DNA fragment analysis, as previously described [30,31].

### 2.4. Treatment and Response

Patient prognosis was estimated with CPSS and CPPS-Mol [10,11] and treatment was based on patient risk group. Treatment response was determined according to the criteria of Savona et al. [32]: no response, complete response, hematological response, partial response, and clinical benefit. Transfusion dependency was defined as needing at least one red blood cell transfusion every 8 weeks over a period of 4 months [11,33].

### 2.5. Statistical Analyses

Categorical variables were summarized as numbers and percentages and continuous variables as median (range). Comparisons between groups were done with the Chi-square test, Fisher’s exact test, Student’s *t*-test, or Mann–Whitney U test, as appropriate. OS was defined as the time from CMML diagnosis to death from any cause or last follow-up. OS after alloSCT was defined as the time from alloSCT to death from any cause or last follow-up. OS was calculated with the Kaplan–Meier method and compared with the log-rank test. Median follow-up was calculated with the Kaplan–Meier estimate of potential follow-up [34]. Time to AML transformation was defined as the time from CMML diagnosis to AML transformation. The cumulative incidence of AML transformation was estimated with a competing risk approach, considering death from any cause without AML transformation as a competing event, and the effect of quantitative covariates was estimated with the Fine-Gray regression model [35]. Non-relapse mortality was defined as the probability of dying after alloSCT without previous occurrence of a relapse, which is a competing event [36]. Patients with no events were censored at the time of last follow-up. All analyses were performed using SPSS version 25 (SPSS Inc., Chicago, IL, USA) and R version 4.2.0. (R Core Team, Vienna, Austria) Two-sided *p*-values < 0.05 were considered statistically significant.

## 3. Results

### 3.1. Patient Characteristics

Baseline patient characteristics are shown in Table 1. Of the 219 patients included in the study, 144 (65.8%) were men and median age at diagnosis was 74.1 years (range, 28–99). Fifty-three patients (24.3%) were transfusion-dependent at diagnosis.

Patients were classified according to cytogenetics, FAB classification [2], WHO categories [1], CPSS [11], and CPSS-Mol [10] when these data were available. Of the 156 patients (71.2%) with informative karyotype results, 106 (67.9%) had a normal karyotype. One-hundred-and-ten patients (70.6%) were classified as cytogenetic low-risk, 23 (14.7%) as intermediate-risk, and 23 (14.7%) as high-risk [9]. Cytogenetic abnormalities are shown in Appendix A. FAB classification identified 148 patients (67.9%) as MDS subtype and 70 (32.1%) as MPN subtype. According to the WHO categories, 151 patients (70.9%) were CMML-0, 35 (16.4%) were CMML-1, and 27 (12.7%) were CMML-2. CPSS classified 57 patients (37.5%) as low, 44 (28.8%) as intermediate-1, 43 (28.1%) as intermediate-2, and nine (5.9%) as high. CPSS-Mol classified 17 (28.3%) as low, 12 (20%) as intermediate-1, 17 (28.3%) as intermediate-2, and 14 (23.3%) as high (Table 1).

### 3.2. Gene Mutations

NGS was performed in 72 patients at diagnosis and in 23 patients at the time of AML transformation. Fourteen patients had available paired NGS data at diagnosis and at AML transformation.

Of the 72 patients with NGS data at diagnosis, 69 (95.8%) had at least one mutation, with a median of three mutations (range, 0–7) per patient. Three patients (4.2%) had no mutations; 13 (18.1%) had one; 18 (25%) had two; 14 (19.4%) had three; and 24 (33.3%) had four or more mutations. (Table 2). The most commonly detected mutation was in *TET2* (59.7%), followed by *SRSF2* (30.2%), *ASXL1* (22.2%), *RUNX1* (20.8%), *CBL* (19.4%), *ZRSR2* (15.9%), *EZH2* (15.9%), and *SETBP1* (14.3%). Other genes were mutated in <10% of patients (Table 2 and Figure 1).

The *TET2* mutation was associated with older age (≥70 years, *p* < 0.001), while *KRAS* and *SETBP1* mutations were associated with younger age (<70 years; *p* = 0.04 and *p* = 0.026, respectively). *TET2* and *ZRSR2* mutations were associated with the MDS phenotype (*p* < 0.001 and *p* = 0.01, respectively) and *ASXL1* and *SETBP1* with the MPN phenotype (*p* = 0.019 and *p* = 0.001, respectively).

### 3.3. Overall Survival

With a median follow-up of 10.2 years (IQR, 5.3–12.3), median OS for the entire cohort was 34 months (95% CI, 28–40) (Figure 2), while it was 28.9 months for patients ≥70 years at diagnosis and 51 months for patients < 70 years (*p* = 0.002) (Table 3).

OS was 51 months for patients with low cytogenetic risk, 30.4 months for those with intermediate risk, and 19.4 months for those with high risk (*p* < 0.001). OS was 40.3 months for patients with the FAB MDS phenotype and 21.3 months for those with the MPN phenotype (*p* = 0.04). OS was 37.7 months for the WHO CMML-0 category, 25 months for the CMML-1 category, and 18.3 months for the CMML-2 category (*p* = 0.227) (Table 3).

OS was 57.2 months for CPSS-defined low-risk patients, 34.8 months for intermediate-1 patients, 19.4 months for intermediate-2 patients, and 13.7 months for high-risk patients (*p* < 0.01) (Table 3 and Figure 3A). According to the dichotomized CPSS categories used in clinical practice (low and intermediate-1 vs. intermediate-2 and high), patients in the low-risk group had an OS of 51.6 months compared to 19.1 months for those in the high-risk group (*p* < 0.01) (Table 3 and Figure 3B). OS was also different according to the CPSS-Mol-defined risk groups but these differences did not reach statistical significance: 85.1 months for low-risk patients; 64.3 months for intermediate-1 patients; 46.8 months for intermediate-2 patients; and 19.1 months for high-risk patients (*p* = 0.09) (Table 3 and Figure 3C). According to the dichotomized CPSS-Mol categories, OS was 67.3 months for the low-risk group and 28.7 months for the high-risk group (*p* = 0.03) (Table 3 and Figure 3D). Nearly half of our patients had a higher individual risk according to CPSS-Mol than according to CPSS. In fact, 25% of patients increased their risk from CPSS low to CPSS-Mol high (Appendix A).

OS was 85.2 months for the 16 patients with fewer than two mutations and 25 months for the 56 with two or more mutations (*p* = 0.006) (Table 3 and Appendix A). The *RUNX1* mutation at CMML diagnosis was associated with shorter OS: 51.1 months for patients with wild-type *RUNX1* vs. 16.6 months for those with the *RUNX1* mutation (*p* = 0.02) (Table 3). Patients harboring at least one mutation in transcription factors (*RUNX1*, *GATA2*, *SETBP1*, *CEBPA*) had an OS of 21.3 months, compared to 64.3 months for those without any of these mutations (*p* = 0.001) (Table 3 and Appendix A). The three patients with the *ASXL1/EZH2* co-mutation had an OS of 2.5 months vs. 35.2 months for those without the co-mutation (*p* = 0.001) (Table 3).

### 3.4. Response to Treatment

Fifty patients (22.8%) received disease-modifying treatment: 34 with only HMA, 13 with only chemotherapy, and 3 with both treatments (Table 1). The median time from diagnosis to initiating disease-modifying treatment was 6.8 months (range 0.1–111.7).

HMA was first implemented in our center in 2011, after which CMML was diagnosed in 96 patients, 37 (38.5%) of whom received HMA and 8 (8.3%) of whom received chemotherapy. Data on response to HMA were available for 36 patients: 10 patients (27.8%) attained a complete response; 6 (16.7%) a partial response; 2 (5.6%) a hematologic response; 1 (2.8%) had clinical benefit; and 17 (47.2%) did not respond. Data on response to chemotherapy was available for 15 patients: eight (53.3%) achieved a complete response; two (13.3%) achieved a partial response; and five (33.3%) did not respond. The *ASXL1* mutation was associated with a lower response rate to chemotherapy: 0% in the two patients with *ASXL1* mutations vs 100% complete response in the five without *ASXL1* mutations (*p* = 0.05).

OS for patients receiving HMA calculated from the start of treatment was 23.8 months (95% CI, 13–34). OS for all 96 patients treated from 2011 on was 35.8 months (95% CI, 22–49), compared to 33 months (95% CI, 24–43) for the 123 patients treated before 2011 (*p* = 0.63).

### 3.5. AlloSCT

Twenty-two patients (10%) received alloSCT, six of whom had progressed to AML at the time of alloSCT. Prior to alloSCT, 9 of these 22 patients (40.9%) had received chemotherapy (3 with chemotherapy plus midostaurin due to *FLT3* mutations), 11 (50%) had received HMA, and 2 (9.1%) had not received any prior treatment. Three patients had received both HMA and chemotherapy prior to alloSCT. Two did not respond to HMA and then received chemotherapy, while the third did not respond to chemotherapy and then received HMA plus venetoclax. Both 2- and 4-year OS after alloSCT were 60.6% (95% CI, 36–78) (Figure 4A). Non-relapse mortality at both 2 and 4 years after alloSCT was 13.6% (95% CI, 3.2–31.4). Ten patients relapsed after alloSCT, with a 2- and 4-year cumulative incidence of relapse after alloSCT of 37.6 % (95% CI, 17–58) and 43.7% (95% CI, 21–64), respectively (Figure 4B). At relapse, 7 of the 10 patients received salvage treatment: 1, donor lymphocyte infusion; 1, enasidenib; 3, sorafenib; 1, HMA plus venetoclax; and 1, chemotherapy. OS calculated from the time of relapse was 15 months (95% CI, 0.03–32).

### 3.6. Targeted Therapies

A total of six patients were treated with targeted therapies. Four patients who relapsed after alloSCT were salvaged with novel targeted therapies: one with enasidenib and three with sorafenib. Three of these patients attained a prolonged complete remission, with a median duration of response of 11 months (95% CI, 8–44). An additional patient received sorafenib post-alloSCT as a preventive measure and a sixth patient received sorafenib at AML transformation.

A 61-year-old man with CMML-2 and an *IDH2* mutation relapsed 4.5 months after alloSCT. He was treated with enasidenib, attained a complete response, and is still in remission after more than 3.5 years. The other five patients received *FLT3* inhibitors (midostaurin or sorafenib). A 55-year-old woman with CMML-2 and *FLT3*-TKD was treated with chemotherapy plus midostaurin followed by alloSCT. She relapsed 2.5 months after alloSCT, immunosuppressive therapy was reduced, and she started treatment with sorafenib. She achieved a complete remission for a year while continuing with sorafenib but died due to non-relapse mortality while still in complete remission. A 56-year-old man with CMML-2 and *TET2, NPM1*, and *FLT3*-ITD (ratio 0.17) mutations received chemotherapy plus midostaurin followed by alloSCT. He relapsed 3 months after alloSCT, immunosuppressive therapy was reduced, and he achieved a complete remission with negative minimal residual disease (MRD). After 6 months, MRD reappeared, he started sorafenib, and he continues to be in complete remission with decreasing MRD kinetics. A 69-year-old man with AML transformation after CMML, an *IDH2* mutation, and *FLT3*-ITD (ratio 0.3) was treated with chemotherapy plus midostaurin followed by alloSCT. Nine months after alloSCT, he relapsed and received salvage treatment: azacytidine plus sorafenib (no response) and decitabine plus venetoclax (no response). He is currently included in a clinical trial of an *IDH2* inhibitor. A 67-year-old man with AML transformation after CMML-2 and *FLT3*-ITD received chemotherapy plus midostaurin followed by alloSCT. He then received maintenance sorafenib for 2 years and is still in complete remission 3 years after alloSCT. Finally, a 69-year-old man with AML transformation after CMML-2, *FLT3*-ITD, and an *IDH2* mutation, who was not eligible for alloSCT, received three different treatments: azacytidine (achieving a partial response), sorafenib plus azacytidine (achieving sustained complete remission for 7 months), and gilteritinib (achieving sustained complete remission for 1 year). However, he then relapsed and died.

### 3.7. AML Transformation

Fifty-three patients (24.2%) progressed to AML, with a median time to transformation of 16.6 months (95% CI, 0.7–124.9). The 2-year and 4-year cumulative incidence of AML transformation were 19% (95% CI, 13.9–24.7%) and 23.3% (95% CI, 17.7–29.4%), respectively (Figure 5A).

When the CPSS score was dichotomized into low and high risk, the 2-year cumulative incidence of AML transformation was 9.6% for low-risk patients vs 39.2% for high-risk patients (*p* < 0.001) (Figure 5B). When the CPSS-Mol score was dichotomized into low and high risk, the 2-year cumulative incidence of AML transformation was 0% for low-risk patients and 25.9% for high-risk patients (*p* < 0.001) (Figure 5C).

At the time of AML transformation, the most commonly detected mutation was in *SRSF2* (47.6%), followed by *TET2* (34.8%), *ASXL1* (30.4%), *IDH2* (26.1%), *RUNX1* (21.7%), *NRAS* (21.7%), *SETBP1* (19%), *DNMT3A* (17.4%), *U2AF1* (14.3%), *SF3B1* (14.3%), *JAK2* (13%), *CBL* (13%), *FLT3* (13%), *ZRSR2* (9.5%), *PHF6* (9.5%), and *KRAS* (8.7%). Other genes were mutated at lower frequencies (Table 2).

The two patients harboring an *IDH2* mutation at CMML diagnosis both progressed to AML and maintained this mutation but with an increased VAF at the time of AML transformation. The two patients with *SF3B1* mutations at diagnosis also progressed to AML. Of the four patients harboring *NPM1* mutations at diagnosis, two progressed to AML and two were treated as AML from the beginning.

Fourteen patients had paired bone marrow samples at CMML diagnosis and at progression to AML (*n* = 13) or to myelofibrosis (*n* = 1) (Figure 6 and Figure 7). Among the patients who progressed to AML, 10 had not previously received disease-modifying treatment and they maintained most of the same mutations with an increased VAF. Six of these 10 patients had also acquired new mutations at the time of AML transformation: *NRAS* (*n* = 2), *FLT3*-ITD (*n* = 2), *FLT3*-TKD (*n* = 1), *PTPN11* (*n* = 1), *RUNX1* (*n* = 2), *ASXL1* (*n* = 1), and *BCOR* (*n* = 1).

Three patients had received disease-modifying treatment prior to AML transformation. One patient maintained the *SETBP1* and *ASXL1* mutations detected at diagnosis and acquired *RUNX1*, *CBL*, and *NF1* mutations. The second patient harbored *BCOR* and *SF3B1* mutations at diagnosis and acquired an *NRAS* mutation at the time of AML transformation. The third patient maintained the *ASXL1*, *SETBP1*, and *SRSF2* mutations detected at diagnosis, but with an increased VAF, and also acquired *RB1*, *ETV6*, and *STAG2* mutations. The patient who progressed to myelofibrosis harbored *SRSF2*, *TET2*, and *MPL* mutations at CMML diagnosis and acquired a *JAK2* mutation at the time of myelofibrosis transformation.

## 4. Discussion

CMML is a rare disease with highly heterogeneous morphological features, clinical manifestations, cytogenetic abnormalities, molecular alterations, and treatment responses. Advances in molecular techniques have made it possible to improve the risk stratification of CMML and to better understand its pathogenesis by taking into account not only cytogenetic abnormalities but also molecular alterations. We retrospectively reviewed the real-world data on 219 CMML patients who were diagnosed and treated at our institution. Our results confirm the poor prognosis of the disease, point out the crucial importance of alloSCT in improving OS, clarify details of clonal evolution to AML, and highlight the feasibility of using new targeted therapies in these patients.

Many of our findings are along the lines of previous reports, including patient characteristics [8,10,11,22,32,37,38,39,40] and median OS (34 months) [8,9,40]. OS was longer in younger patients (*p* = 0.002), those with low cytogenetic risk (*p* < 0.001), and those with the FAB MDS phenotype (*p* = 0.04), as previously reported [8,40]. Also in line with other studies [41], differences in OS according to the current WHO classification (CMML-0/CMML-1/2) were not significant.

The mutational landscape of our patients was also similar to that of previous studies [7,15,42,43]. For example, the vast majority of our patients harbored at least one somatic mutation. While the most frequently mutated gene in previous studies was *TET2*, followed by *SRSF2*, *ASXL1*, and the *RAS* pathway genes (*NRAS, KRAS*, and *CBL*) [15,42], in our study, the most frequently mutated genes were *TET2, SRSF2, ASXL1, RUNX1*, and *CBL*. In line with a previous study [15], we identified a higher incidence of *TET2* mutations in patients ≥70 years, suggesting that older age is related to a higher frequency of clonal hematopoiesis [44]. In contrast, *KRAS* and *SETBP1* mutations were more frequent in younger patients.

Although several studies have explored the clinical impact of mutations in CMML, the issue remains unresolved. In the present study, patients with fewer than two gene mutations at CMML diagnosis had longer survival (*p* = 0.006), which is in line with some previous reports [10,15]. The negative prognostic impact of *ASXL1, NRAS, SETBP1*, and/or *RUNX1* mutations has been incorporated in the CPSS-Mol prognostic score [7,42,45], and we found that half of our patients were classified as higher risk by CPSS-Mol than by CPSS. In fact, CPSS-Mol may be a superior method for calculating risk and deciding on treatment, as patients classified as low-risk by CPSS may well benefit from the treatment that would be indicated by CPSS-Mol. In line with previous reports, the *EZH2/ASXL1* co-mutation [22,46,47], present in three of our patients, and *RUNX1* mutations [10], detected in 20.8% of our patients, were also associated with shorter OS (*p* = 0.001 and *p* = 0.02, respectively). Interestingly, patients harboring a mutation in at least one of the transcription factors (*RUNX1*, *GATA2*, *SETBP1*, and *CEBPA*) had shorter OS (*p* = 0.001); to the best of our knowledge, this phenomenon has not been explored in previous studies.

The MDS phenotype of CMML has been associated with mutations of *TET2, U2AF1*, and *SF3B1* [48], and the MPN phenotype with mutations that activate *RAS* family members, *ASXL1, JAK2, SETBP1, SRSF2*, and *EZH2* [10,22,46]. In our patients, *TET*2 and *ZRSR2* mutations were associated with the MDS phenotype and *ASXL1* and *SETBP1* with the MPN phenotype.

The percentage of patients who progressed to AML in our study was also along the lines of previous reports [11,22,47,49]. The risk of AML transformation depends on the transformative aggressiveness of CMML and on the competing risk of dying before transformation, usually from comorbidities or consequences of cytopenia [37]. Findings in our cohort confirm the capacity of dichotomized CPPS and CPSS-Mol scores to discriminate between patients with low and high risk of AML transformation [10,11]. The most frequently mutated genes at the time of AML transformation in our cohort were *SRSF2, TET2, ASXL1, IDH2, RUNX1, NRAS, SETBP1*, and *DNMT3A*.

Comprehensive information about clonal evolution from CMML to AML is still lacking. A few sequential studies comparing paired patient samples at the time of CMML diagnosis and at AML transformation have postulated a key role for the *RAS* pathway and for mutations in *EZH2, IDH1/2, NPM1*, and *FLT3*-ITD [14,50]. Patel et al. [16] suggested that *TET2, ASXL1*, and *SRSF2* were ancestral mutations present in early stages of CMML and that *KRAS, NRAS, RUNX1, U2AF1*, and *CBL* mutations were secondary events leading to AML transformation. In our study, we observed newly acquired mutations at AML transformation, including in genes involved in signaling pathways (*NRAS, FLT3, JAK2, PTPN11*, and *CBL*), as well as in *RUNX1* and *ASXL1*. While we cannot draw definitive conclusions from our findings in only 14 patients, we can speculate that, in line with the suggestion by Patel et al. [16], there is a connection between the acquisition of new mutations and AML transformation. If this hypothesis is borne out in further studies, we can recommend serial analyses during follow-up of CMML patients to predict progression to AML and adapt treatment accordingly. Importantly, *FLT3* mutations, which are druggable, appeared at AML transformation in two patients.

Interestingly, the two patients in our cohort who had an *IDH2* mutation at diagnosis and two of the four patients with mutated *NPM1* at diagnosis progressed to AML, while the other two patients with *NPM1* mutations received chemotherapy followed by alloSCT despite having <20% blasts, following previous recommendations [51] that prognosis is improved with more intense therapy.

Only a quarter of our patients diagnosed from 2011 that received disease-modifying treatment (HMAs or chemotherapy) and less than 50% of these achieved any type of response. These findings are in line with other studies. A large retrospective non-randomized study indicated that HMAs conferred a survival advantage over hydroxyurea in high-risk, but not low-risk, CMML patients [38]. In contrast, the DACOTA trial (EudraCT 2014-000200-10) [52] failed to demonstrate a benefit for the HMA decitabine over hydroxyurea in proliferative CMML. HMAs in combination with venetoclax have improved outcomes in AML patients who are ineligible for intense chemotherapy [53]. However, their role in CMML and AML transformation remains unclear, though they have been proposed as a feasible and effective bridge to alloSCT [25,26].

In fact, no available drug has been able to improve the natural course of CMML and alloSCT remains the only potentially curative therapy. In our cohort, only 22 patients (10%) underwent alloSCT, which is similar to previous reports [18,54], mainly because the majority of CMML patients are elderly and not eligible for alloSCT. However, the outcomes of our patients after alloSCT were slightly better than those of other studies [55,56]. Our patients achieved a 4-year OS of 61%, and many of our patients were long survivors despite a cumulative incidence of relapse at four years of 43.7%. This may be due, at least in part, to the fact that three patients who relapsed after alloSCT and were salvaged with new targeted therapies attained a prolonged complete remission, with a median duration of response of 11 months. In the last few years, several targeted therapies have been approved for treatment of other myeloid diseases, such as AML, but their efficacy in CMML is largely unknown [25,26,27,28]. In fact, six of our patients harbored *FLT3* or *IDH* mutations, which are druggable, and after treatment with *FLT3* and *IDH* inhibitors, some attained a long-lasting response. These findings indicate that novel targeted therapies are feasible in CMML and suggest that new drug combinations could induce quicker and deeper responses and thus improve alloSCT outcomes [26,55]. In this line, there are several targeted drugs on an early phase of development (preclinical models or clinical trials) being tested in myeloid pathology that could be of utility on CMML in a foreseeable future. Some examples includes the orally bioavailable modulator of the SF3b complex H3B-8800 for patients with splicing mutations as *SRSF2* [57,58,59,60]; MEK kinase inhibitors, for patients harboring a *RAS* Pathway-Activating Mutation [61,62,63]; or cotreatment with Omacetaxine (an inhibitors of protein translation) and a *BCL2* inhibitor for targeting *RUNX1* mutations (preclinical model study) [64]. Further, they highlight the importance of exhaustive molecular characterization at diagnosis and at progression. NGS is an extremely useful tool that makes it feasible to perform this molecular characterization. It can provide valuable information on gene alterations that can help determine the diagnosis and prognosis of CMML patients, improve patient risk stratification, use available treatments more rationally, and identify potential therapeutic targets.

The present study has several limitations, including its retrospective nature and the limited number of patients in whom NGS was performed. However, since CMML patients are generally not included in clinical trials because of the low frequency and heterogeneity of the disease, at present we must rely on real-world data for information on prognosis, molecular characterization, and treatment response. However, the DACOTA trial [52] proved that randomized clinical trials are feasible in CMML, and clinical trials in CMML are now ongoing. Enrollment in CMML-specific clinical trials or in exploratory trials of specific tumor biomarkers should be encouraged as the best way to clarify the molecular mechanisms behind CMML and to explore the efficacy of new drugs in this disease [22,24,47,65,66]. Nevertheless, until such time where we have reliable data from clinical trials, studies such as ours, based on real-world data, can provide valuable insight into the CMML mutational landscape and treatment outcomes.

## 5. Conclusions

Our comprehensive clinical and molecular characterization of a large cohort of CMML patients shows that despite advances in risk stratification and the increasing use of HMAs, the prognosis of CMML continues to be poor. AlloSCT is the only potentially curative treatment, but only a small proportion of CMML patients are eligible candidates, mainly due to older age and comorbidities. There is clearly an unmet need for improved treatment for these patients. Complete molecular characterization can help us to understand the pathophysiology of CMML, to improve risk stratification, and to better understand transformation to AML. The incorporation of novel therapies targeting specific gene alterations could contribute to a more precise treatment, with a higher response rate, lower toxicity, and greater survival benefit. Since CMML-specific clinical trials are difficult to perform due to the low frequency and the heterogeneity of the disease and the age of the patients, more studies of real-world data similar to the present one can provide information on prognosis in CMML.

## Figures and Tables

**Figure 1 cancers-14-04107-f001:**
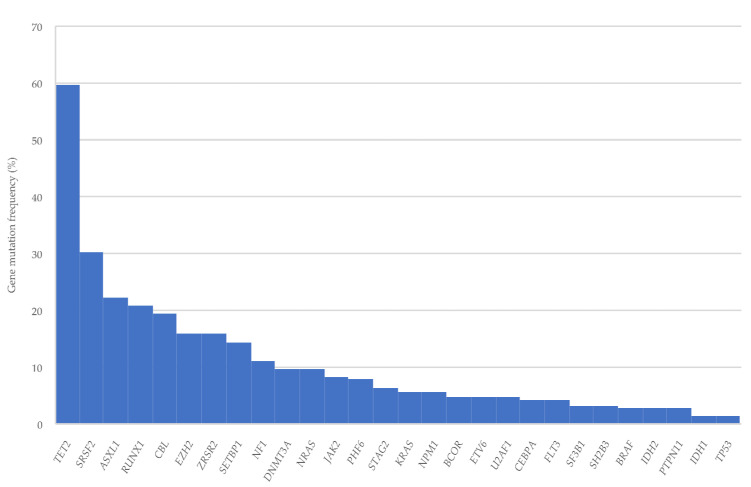
Gene mutations detected at CMML diagnosis.

**Figure 2 cancers-14-04107-f002:**
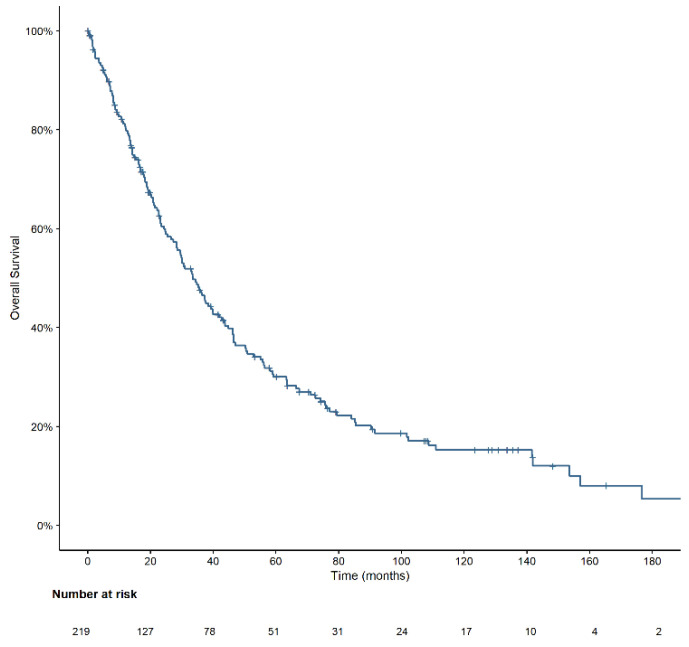
Overall survival of the entire cohort of CMML patients (*n* = 219).

**Figure 3 cancers-14-04107-f003:**
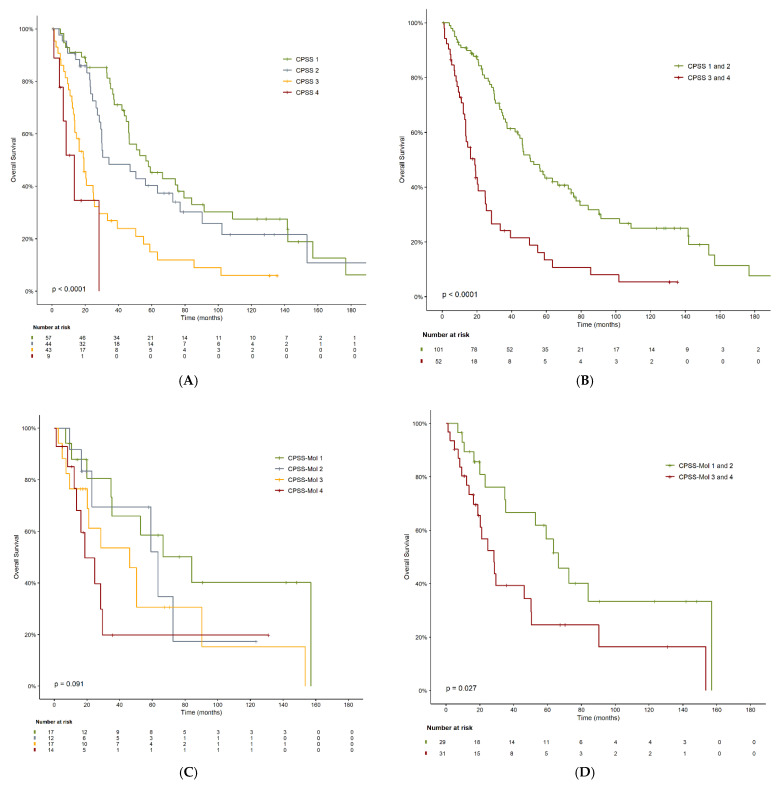
Overall survival according to (**A**) CPSS risk score; (**B**) CPSS risk score dichotomized into low risk (low and intermediate-1) and high risk (intermediate-2 and high); (**C**) CPSS-Mol risk score; and (**D**) CPSS-Mol dichotomized into low risk (low and intermediate-1) and high risk (intermediate-2 and high).

**Figure 4 cancers-14-04107-f004:**
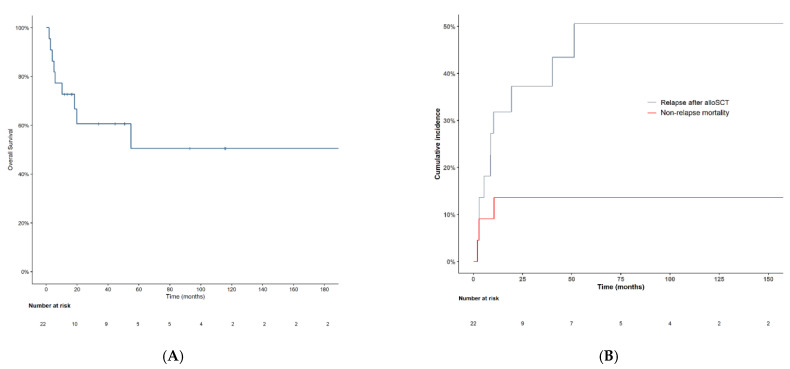
(**A**) Overall survival and (**B**) cumulative incidence of relapse after alloSCT.

**Figure 5 cancers-14-04107-f005:**
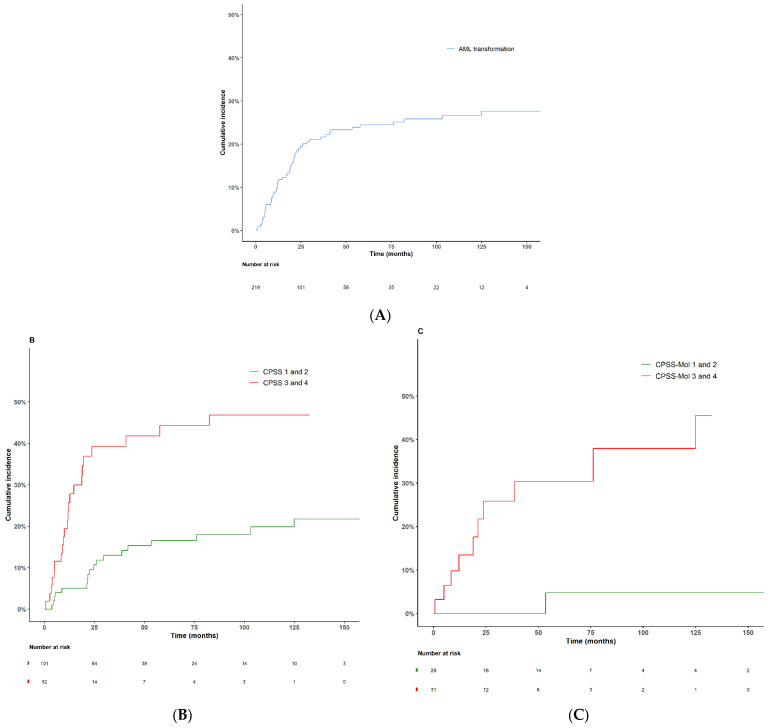
Cumulative incidence of transformation to AML (**A**) for the entire cohort (*n* = 219), (**B**) according to CPSS prognostic score, and (**C**) according to CPSS-Mol prognostic score.

**Figure 6 cancers-14-04107-f006:**
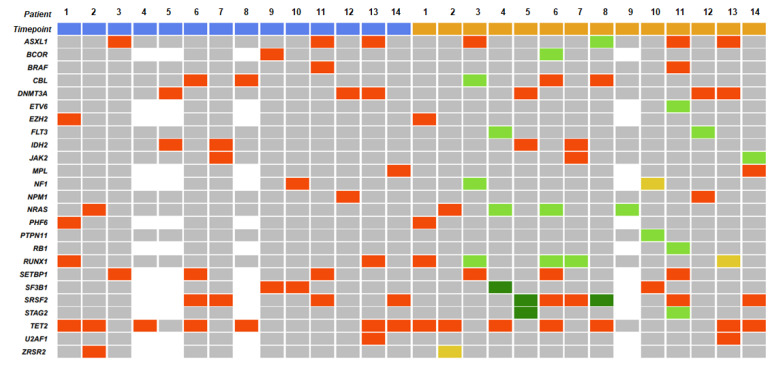
Mutational landscape in 14 CMML patients who transformed to AML (*n* = 13) or to myelofibrosis (*n* = 1). Each row corresponds to one gene. Each blue square corresponds to one patient at CMML diagnosis, and each orange square corresponds to one patient at transformation. White squares indicate unanalyzed genes. Gray squares indicate unmutated genes. Red squares indicate that the original mutation was retained at transformation. Light green squares indicate mutations acquired at transformation. Yellow squares indicate mutations not retained at transformation. Dark green squares indicate a mutation present at transformation but not analyzed at diagnosis. Ten patients (1, 2, 4, 5, 6, 7, 8, 10, 12, and 13) had not previously received disease-modifying treatment and they maintained most of the same mutations with an increased VAF. Patients 4, 6, 7, 8, 10, and 12 had acquired new mutations at the time of AML transformation. Three patients (3, 9, and 11) had received disease-modifying treatment prior to AML transformation.

**Figure 7 cancers-14-04107-f007:**
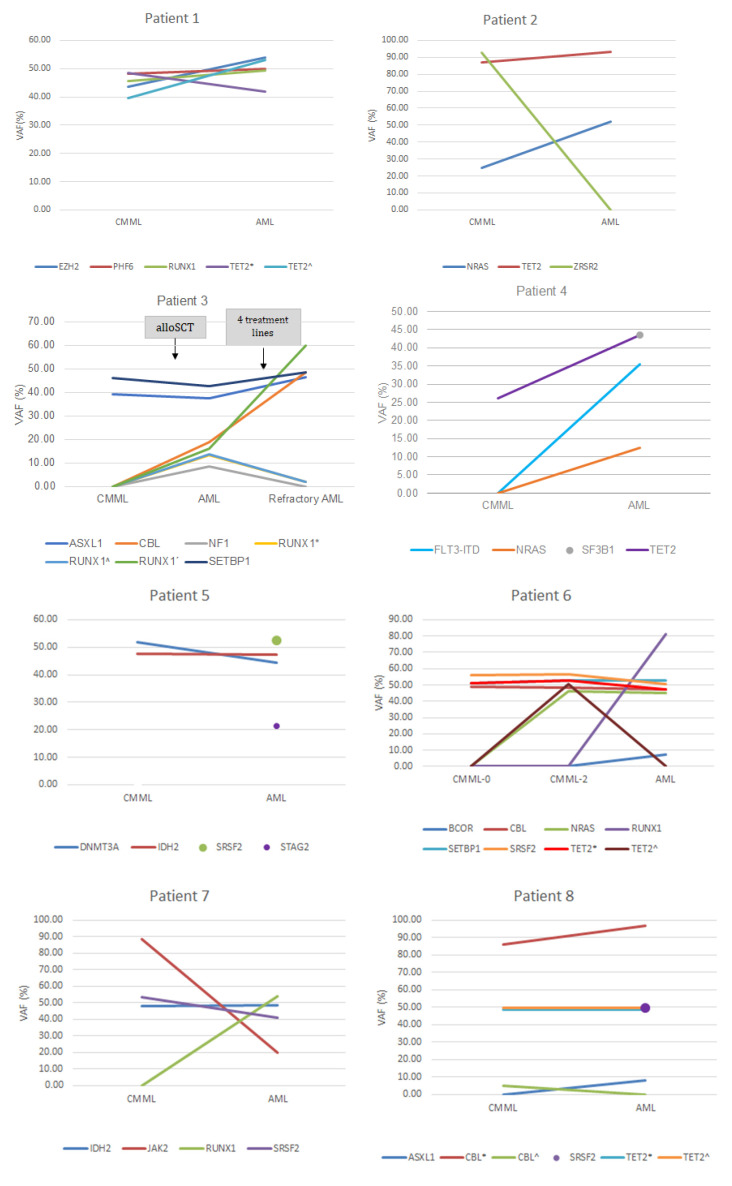
Variant allele frequency (VAF) plots of mutated genes in 14 CMML patients who transformed to AML (*n* = 13) or to myelofibrosis (*n* = 1). Disease-modifying treatments are shown. Symbols are used to differentiate variants of mutations (*, ^, ´ and ¨).

**Table 1 cancers-14-04107-t001:** Clinical characteristics of CMML patients.

Characteristics at Diagnosis	Patients (*n* = 219)
Age, years—median (range)	74.1 (28–99)
Men—*n* (%)	144 (65.8)
WBC count—median (range)	9.47 (3–119) × 10^9^/L
Neutrophil count—median (range)	4.7 (0.49–57.12) × 10^9^/L
Monocyte count—median (range)	1.91 (1–33.72) × 10^9^/L
% monocytes—median (range)	20 (10–60)
Platelets—median (range)	126 (7–1037) × 10^9^/L
Hemoglobin—median (range)	11.6 (4.9–16.1) g/dl
Blast percentage ^1^—median (range)	3% (0–19)
Transfusion dependence—*n* (%)	53 (24.3)
Presentation—therapy related *n* (%)	14 (7.3)
Cytogenetic risk—*n* (%)	*n* = 156
low	110 (70.6)
intermediate	23 (14.7)
high	23 (14.7)
Cytogenetic profiling—*n* (%)	*n* = 156
normal	106 (67.9)
abnormal	50 (32.1)
FAB classification—*n* (%)	*n* = 218
myelodysplastic	148 (67.9)
myeloproliferative	70 (32.1)
WHO classification—*n* (%)	*n* = 213
CMML-0	151 (70.9)
CMML-1	35 (16.4)
CMML-2	27 (12.7)
CPSS classification—*n* (%)	*n* = 153
low	57 (37.5)
intermediate-1	44 (28.8)
intermediate-2	43 (28.1)
high	9 (5.9)
CPSS-Mol classification—*n* (%)	*n* = 60
low	17 (28.3)
intermediate-1	12 (20)
intermediate-2	17 (28.3)
high	14 (23.3)
Patients receiving disease-modifying treatment—*n* (%) ^2^	*n* = 50
Only HMA	34 (68)
Only chemotherapy	13 (26)
Both HMA and chemotherapy	3 (6)
Patients receiving alloSCT—*n* (%)	22 (10)
Patients progressing to AML—*n* (%)	53 (24.2)

^1^ Bone marrow blasts. ^2^ The first patient was treated with HMA in 2011. From then on, 96 patients were diagnosed with CMML, 37 (38.54%) of whom received HMA. Of the three patients who received both treatments, two received first HMA and then chemotherapy, and the other received first chemotherapy and then HMA.

**Table 2 cancers-14-04107-t002:** Gene mutations in CMML patients.

Gene Mutations	At CMML Diagnosis	At AML Transformation
	Patients (*n* = 72) *n* (*n*/N, %) ^1^	Patients (*n* = 23) *n* (*n*/N, %) ^1^
Number of mutated genes per patient		
0	3 (3/72, 4.2)	1 (1/23, 4.3)
1	13 (13/72, 18.1)	1 (1/23, 4.3)
2	18 (18/72, 25)	3 (3/23, 13)
3	14 (14/72, 19.4)	3 (3/23, 13)
≥4	24 (24/72, 33.3)	15 (15/23, 65.1)
Most frequently mutated genes		
*TET2*	43 (43/72, 59.7)	8 (8/23, 34.8)
*SRSF2*	19 (19/63, 30.2)	10 (10/21, 47.6)
*ASXL1*	16 (16/72, 22.2)	7 (7/23, 30.4)
*RUNX1*	15 (15/72, 20.8)	5 (5/23, 21.7)
*CBL*	14 (14/72, 19.4)	3 (3/23, 13)
*ZRSR2*	10 (10/63, 15.9)	2 (2/21, 9.5)
*EZH2*	10 (10/63, 15.9)	1 (1/21, 4.8)
*SETBP1*	9 (9/63, 14.3)	4 (4/21, 19)
*IDH2*	2 (2/72, 2.8)	6 (6/23, 26.1)
*NRAS*	7 (7/72, 9.7)	5 (5/23, 21.7)
*DNMT3A*	7 (7/72, 9.7)	4 (4/23, 17.4)
*U2AF1*	3 (3/63, 4.8)	3 (3/21, 14.3)
*SF3B1*	2 (2/63, 3.2)	3 (3/21, 14.3)
*JAK2*	6 (6/72, 8.3)	3 (3/23, 13)
*FLT3*	3 (3/72, 4.2)	3 (3/23, 13)
*PHF6*	5 (5/63, 7.9)	2 (2/21, 9.5)
*KRAS*	4 (4/72, 5.6)	2 (2/23, 8.7)
Genes in signaling pathways		
*NRAS*	7 (7/72, 9.7)	5 (5/23, 21.7)
*KRAS*	4 (4/72, 5.6)	2 (2/23, 8.7)
*FLT3*	3 (3/72, 4.2)	3 (3/23, 13)
*CSF3R*	0 (0/63, 0)	1 (1/21, 4.8)
*JAK2*	6 (6/72, 8.3)	3 (3/23, 13)
*CBL*	14 (14/72, 19.4)	3 (3/23, 13)
*PTPN11*	2 (2/71, 2.8)	1 (1/23, 4.3)
Epigenetic regulators		
*TET2*	43 (43/72, 59.7)	8 (8/23, 34.8)
*IDH2*	2 (2/72, 2.8)	6 (6/23, 26.1)
*DNMT3A*	7 (7/72, 9.7)	4 (4/23, 17.4)
*ASXL1*	16 (16/72, 22.2)	7 (7/23, 30.4)
*EZH2*	10 (10/63, 15.9)	1 (1/21, 4.8)
*IDH1*	1 (1/72, 1.4)	1 (1/23, 4.3)
Transcription factors		
*RUNX1*	15 (15/72, 20.8)	5 (5/23, 21.7)
*SETBP1*	9 (9/63, 14.3)	4 (4/21, 19)
*CEBPA*	3 (3/72, 4.2)	0 (0/23, 0)
Spliceosome complex		
*SRSF2*	19 (19/63, 30.2)	10 (10/21, 47.6)
*SF3B1*	2 (2/63, 3.2)	3 (3/21, 14.3)
*ZRSR2*	10 (10/63, 15.9)	2 (2/21, 9.5)
*U2AF1*	3 (3/63, 4.8)	3 (3/21, 14.3)
DNA damage response genes		
*TP53*	1 (1/72, 1.4)	1 (1/23, 4.3)
*PHF6*	5 (5/63, 7.9)	2 (2/21, 9.5)
Others		
*NPM1*	4 (4/72, 5.6)	2 (2/23, 8.7)

^1^ *n*, number of patients with the mutation; *N*, number of patients with informative NGS results.

**Table 3 cancers-14-04107-t003:** Overall survival according to age, risk categories, and gene mutations.

Factor	No. Patients	OS (Months)	95% CI	*p*
**Age**				0.002
≥70 years	142	28.9	21.0–37.3	
<70 years	77	51.0	37.9–64.1	
**Cytogenetic risk**				0.001
Low	110	51.0	40.7–61.4	
Intermediate	22	30.4	23.6–37.2	
High	24	19.4	9.1–29.8	
**FAB**				0.04
MDS	148	40.3	32.3–49.3	
MPN	70	21.3	14.7–27.8	
**WHO**				0.227
CMML-0	151	37.7	29.4–45.9	
CMML-1	35	25.0	19.1–30.9	
CMML-2	27	18.3	11.3–25.4	
**CPSS**				0.01
Low	57	57.2	35.4–78.8	
Intermediate-1	44	34.8	10.3–59.3	
Intermediate-2	43	19.4	13.8–25.2	
High	9	13.7	5.6–21.8	
**Dichotomized CPSS**				0.01
Low	101	51.6	39.9–63.2	
High	52	19.1	13.3–24.9	
**CPSS-Mol**				0.09
Low	17	85.1	37.6–132.7	
Intermediate-1	12	64.3	20.3–108.3	
Intermediate-2	17	46.8	12.8–80.8	
High	14	19.1	6.7–31.5	
**Dichotomized CPSS-Mol**				0.03
Low	29	67.3	49.6–85.1	
High	31	28.7	17.2–40.0	
**No. gene mutations**				0.006
<2 mutations	16	85.2	51.3–119.0	
≥2 mutations	56	25.0	17.6–32.5	
**RUNX1 mutation**				0.02
Detected	15	16.6	10.0–23.0	
Not detected	57	51.1	21.5–80.8	
**Mutations in transcription factors**				0.001
≥1 mutation detected	25	21.3	12.0–30.4	
No mutation detected	41	64.3	31.8–96.9	
**ASXL1/EZH2 co-mutation**				0.001
Detected	3	2.5	0.3–4.7	
Not detected	68	35.2	10.3–60.2	

## Data Availability

Due to privacy and ethical concerns, the data that support the findings of this study are available on request from the corresponding author.

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
