# Peer review of "Real-World Data on Chronic Myelomonocytic Leukemia: Clinical and Molecular Characteristics, Treatment, Emerging Drugs, and Patient Outcomes"

_cancers, 2022, doi:10.3390/cancers14174107_

Round 1

Reviewer 1 Report

The authors have presented a real-world study to distinguish the mutations and the potential treatment targets for prevention of transformation from CMML to AML. Overall, the study was critically designed and well performed. Data were pretty clear presented and the study results supported the main statement. 

One suggestion, in the discussion part, if there could be some more information about the concurrent drugs or candidates that are targeting to the most commonly detected mutations such as TET2, SRSF2, ASXL1 and RUNX1, it should be more attractive to a wider readers.

Author Response

Response 1: Dear Reviewer #1, thank you for your constructive suggestion. As recommendend, we have included in the discussion additional data about candidates drugs targeting some of the most commonly detected mutations in CMML:

“In this line, there are several targeted drugs on an early phase of development (preclinical models or clinical trials) being tested in myeloid pathology that could be of utility on CMML in a forseable future. Some examples includes the orally bioavailable modulator of the SF3b complex H3B-8800 for patients with splicing mutations as SRSF2 [57-60]; MEK kinase inhibitors, for patients harboring a RAS Pathway-Activating Mutation [61-63]; or cotreatment with Omacetaxine (an inhibitor of protein translation) and a BCL2 inhibithor for targeting RUNX1 mutations (preclinical model study) [64]”.

References:

  1. Steensma, D.P.; Wermke, M.; Klimek, V.M.; Greenberg, P.L.; Font, P.; Komrokji, R.S.; Yang, J.; Brunner, A.M.; Carraway, H.E.; Ades, L.; Al-Kali, A.; Alonso-Dominguez, J.M.; Alfonso-Pierola, A.; Coombs, C.C.; Deeg, H.J.; Flinn, I.; Foran, J.M.; Garcia-Manero, G.; Maris, M.B.; McMasters, M.; Micol, J.B.; De Oteyza, J.P.; Thol, F.; Wang, E.S.; Watts, J.M.; Taylor, J.; Stone, R.; Gourineni, V.; Marino, A.J.; Yao, H.; Destenaves, B.; Yuan, X.; Yu, K.; Dar, S.; Ohanjanian, L.; Kuida, K.; Xiao, J.; Scholz, C.; Gualberto, A.; Platzbecker, U. Phase I First-in-Human Dose Escalation Study of the oral SF3B1 modulator H3B-8800 in myeloid neoplasms. Leukemia 2021,35, pp 3542-3550. https://doi.org/10.1038/s41375-021-01328-9
  2. Rioux, N.; Smith, S.; Colombo, F.; Kim, A.; Lai, W.G.; Nix, D.; Siu, Y.A.; Schindler, J.; Smith, P.G. Metabolic disposition of H3B-8800, an orally available small-molecule splicing modulator, in rats, monkeys, and humans. Xenobiotica 2020,50, pp 1101-1114. https://doi.org/10.1080/00498254.2019.1709134
  3. Steensma, D.P.; Wermke, M.; Klimek, V.M.; Greenberg, P.L.; Font, P.; Komrokji, R.S.; Yang, J.; Brunner, A.M.; Carraway, H.E.; Ades, L.; Al-Kali, A.; Alonso Dominguez, J.M.; Alonso, A.; Coombs, C.C.; Deeg, H.J.; Donnellan, W.B.; Foran, J.M.; Garcia-Manero, G.; Maris, M.B.; McMasters, M.; Micol, J.-B.; Perez De Oteyza, J.; Thol, F.; Wang, E.S.; Watts, J.M.; Buonamici, S.; Kim, A.; Gourineni, V.; Marino, A.J.; Rioux, N.; Schindler, J.; Smith, S.; Yao, H.; Yuan, X.; Yu, K.; Platzbecker, U. Results of a Clinical Trial of H3B-8800, a Splicing Modulator, in Patients with Myelodysplastic Syndromes (MDS), Acute Myeloid Leukemia (AML) or Chronic Myelomonocytic Leukemia (CMML). Blood 2019,134, pp 673-673. https://doi.org/10.1182/blood-2019-123854
  4. Seiler, M.; Yoshimi, A.; Darman, R.; Chan, B.; Keaney, G.; Thomas, M.; Agrawal, A.A.; Caleb, B.; Csibi, A.; Sean, E.; Fekkes, P.; Karr, C.; Klimek, V.; Lai, G.; Lee, L.; Kumar, P.; Lee, S.C.; Liu, X.; Mackenzie, C.; Meeske, C.; Mizui, Y.; Padron, E.; Park, E.; Pazolli, E.; Peng, S.; Prajapati, S.; Taylor, J.; Teng, T.; Wang, J.; Warmuth, M.; Yao, H.; Yu, L.; Zhu, P.; Abdel-Wahab, O.; Smith, P.G.; Buonamici, S. H3B-8800, an orally available small-molecule splicing modulator, induces lethality in spliceosome-mutant cancers. Nat Med 2018,24, pp 497-504. https://www.ncbi.nlm.nih.gov/pubmed/29457796
  5. Kloos, A.; Mintzas, K.; Winckler, L.; Gabdoulline, R.; Alwie, Y.; Jyotsana, N.; Kattre, N.; Schottmann, R.; Scherr, M.; Gupta, C.; Adams, F.F.; Schwarzer, A.; Heckl, D.; Schambach, A.; Imren, S.; Humphries, R.K.; Ganser, A.; Thol, F.; Heuser, M. Effective drug treatment identified by in vivo screening in a transplantable patient-derived xenograft model of chronic myelomonocytic leukemia. Leukemia 2020,34, pp 2951-2963. https://www.ncbi.nlm.nih.gov/pubmed/32576961
  6. Desikan, S.P.; Ravandi, F.; Pemmaraju, N.; Konopleva, M.; Loghavi, S.; Borthakur, G.; Jabbour, E.J.; Daver, N.; Jain, N.; Chien, K.S.; Maiti, A.; Montalban-Bravo, G.; Kadia, T.M.; Kwari, M.; Kantarjian, H.; Short, N.J. A Phase II Study of Azacitidine, Venetoclax and Trametinib in Relapsed/Refractory AML Harboring a Ras Pathway-Activating Mutation. Blood 2021,138, pp 4436-4436. https://doi.org/10.1182/blood-2021-151707
  7. Borthakur, G.; Popplewell, L.; Boyiadzis, M.; Foran, J.; Platzbecker, U.; Vey, N.; Walter, R.B.; Olin, R.; Raza, A.; Giagounidis, A.; Al-Kali, A.; Jabbour, E.; Kadia, T.; Garcia-Manero, G.; Bauman, J.W.; Wu, Y.; Liu, Y.; Schramek, D.; Cox, D.S.; Wissel, P.; Kantarjian, H. Activity of the oral mitogen-activated protein kinase kinase inhibitor trametinib in RAS-mutant relapsed or refractory myeloid malignancies. Cancer 2016,122, pp 1871-1879. https://www.ncbi.nlm.nih.gov/pubmed/26990290
  8. Mill, C.P.e.a. Effective therapy for AML with RUNX1 mutation by cotreatment with inhibitors of protein translation and BCL2. Blood 2022,139, pp 907-921. https://doi.org/10.1182/blood.2021013156

Reviewer 2 Report

The paper is an excellent example of real-world data on a large cohort of CMML patients. From my point of view, the paper is well structured. The results are described in an orderly and easy-to-read manner. The discussion is complete, comparing the results obtained with other works, and emphasizing the limitations of the study.

I just have one small comment. The authors define non-relapse mortality as a time variable (time since alloSCT to death from any cause in the absence of previous disease persistence or recurrence). In the results, however, they talk about % non-relapse mortality at both 2 and 4 years. Is it really a time-dependent variable?

In the other hand, just a small typo. Line 240, OS: 51.1 months, and in table 3 the decimal is missing (51).

Thank you very much! It is an excellent job. 

Author Response

Response 1: Dear reviewer #2, thanks for your positive comments and for pointing out the misinterpretation in the definition of non-relapse mortality. We agree with this reviewer that the definition of the non-relapse mortality need to be clarified since we are really calculating the cumulative incidence that is showing the probability of dying without previous occurrence of a relapse, which is a competitive event. Therefore, to clarify this we have corrected the definition of non-relapse mortality in Methods section as follows:

“Non-relapse mortality was defined as the probability of dying after alloSCT without previous occurrence of a relapse which is a competing event [36].”

In the other hand, just a small typo. Line 240, OS: 51.1 months, and in table 3 the decimal is missing (51).

Response 2: Following the reviewer’s recommendation we added the decimals in table 3.

Reference:

  1. Iacobelli, S.; Committee, E.S. Suggestions on the use of statistical methodologies in studies of the European Group for Blood and Marrow Transplantation. Bone Marrow Transplant 2013,48 Suppl 1, pp S1-37. https://www.ncbi.nlm.nih.gov/pubmed/23462821
